# Transmission, Strain Diversity, and Zoonotic Potential of Chronic Wasting Disease

**DOI:** 10.3390/v14071390

**Published:** 2022-06-25

**Authors:** Sandra Pritzkow

**Affiliations:** Mitchell Center for Alzheimer’s Disease and Related Brain Disorders, Department of Neurology, University of Texas McGovern Medical School, Houston, TX 77030, USA; sandra.pritzkow@uth.tmc.edu

**Keywords:** prions, prion diseases, chronic wasting disease, prion strains, PMCA, spillover potential, zoonotic potential

## Abstract

Chronic wasting disease (CWD) is a prion disease affecting several species of captive and free-ranging cervids. In the past few decades, CWD has been spreading uncontrollably, mostly in North America, resulting in a high increase of CWD incidence but also a substantially higher number of geographical regions affected. The massive increase in CWD poses risks at several levels, including contamination of the environment, transmission to animals cohabiting with cervids, and more importantly, a putative transmission to humans. In this review, I will describe the mechanisms and routes responsible for the efficient transmission of CWD, the strain diversity of natural CWD, its spillover and zoonotic potential and strategies to minimize the CWD threat.

## 1. Background

Prion diseases, also known as Transmissible spongiform encephalopathies (TSEs), are a group of fatal, infectious, neurodegenerative disorders of the brain affecting humans and several other species of mammals, including sheep, goats, cattle, cervids, mink, felines, and camelids [1]. In humans, TSEs are exceedingly rare. Even the most common human TSE is still very infrequent, as is the case of Creutzfeldt–Jakob disease (CJD) which appears at an average rate of 1 new case per million people per year [1,2]. In animals, the most common TSEs are sheep scrapie, bovine spongiform encephalopathy (BSE) affecting cattle, and chronic wasting disease (CWD) affecting captive and wild cervids [2].

The infectious agent in prion disease, usually referred as prion, is likely composed exclusively by a misfolded form of the prion protein (PrP^Sc^), which has the uncanny ability to convert the natively folded prion protein (PrP^C^) naturally present in many cell types, mostly neurons [3,4]. Molecularly, PrP^Sc^ is an aggregate of variable size adopting an intermolecular β-sheet-rich structure [4]. The aggregate acts as a seed to capture the mostly α-helical PrP^C^, which becomes incorporated into the aggregate adopting the same misfolded structure as the parent PrP^Sc^ [5]. Strikingly, this protein-based infectious agent satisfies the Koch postulates used to define a causative relationship between a microbe and a disease [5,6]. Indeed, prions can faithfully multiply in an appropriate host, be transmitted among individuals by various routes including foodborne and bloodborne, can be titrated by infectivity bioassays and display strain diversity, transmission controlled by species barrier, and the ability to cross biological membranes [5].

Among TSEs, CWD is perhaps the most prevalent and worrisome member of the group. It is so far the only prion disease in wildlife, affecting various species of cervid, including white-tailed deer (*Odocoileus virginianus*), mule deer (*Odocoileus hemionus*), reindeer (*Rangifer tarandus*), red deer (*Cervus elaphus*), elk (*Cervus canadensis*), moose (*Alces alces*), sika (*Cervus nippon*), and muntjac (*Muntiacus muntjak*) [7]. CWD is very contagious and its origin, prevalence, and mechanisms of transmission are not entirely understood [7]. Symptoms of late-stage CWD infection include emaciation, ataxia, excessive salivation, depression, muscle wasting, and weakness [7,8,9]. It has been shown that CWD produces a significant decline of the wild deer population in areas with high incidence to CWD [10]. CWD-affected deer were 4.5 times more likely to die per year than noninfected deer [10].

CWD was first reported in 1967 in a captive deer facility in Colorado [8]. The origin of CWD is unknown [11], but several possibilities have been suggested (Figure 1). It has been proposed that CWD emerged due to infection with scrapie from sheep that commonly cohabit with cervids [12,13]. Supporting this conclusion, experimental infection of cervids with scrapie-contaminated brain homogenate showed that scrapie prions from sheep can infect elk and white-tailed deer [13]. Moreover, studies by Greenlee and collaborators showed that experimental infection of sheep with CWD produce a disease very similar to conventional scrapie [14], providing further evidence for a possible origin of CWD from scrapie. Other alternative for CWD origin is the appearance of mutations in the prion protein gene (*Prnp*) which resulted in a protein more prone to convert to PrP^Sc^ (Figure 1) [7]. In humans, up to 15% of the TSE cases arise from *Prnp* mutations [1]. In animals, a single mutation led to the emergence of one of the BSE cases in USA [15]. Another alternative is a spontaneous switch on PrP^C^ conformation (Figure 1), perhaps triggered by changes on the cellular milieu or a defect on the cellular response to protein misfolding [7].

In the past several decades, the disease has been rapidly and progressively spreading geographically and now affects 30 states in the USA, four Canadian provinces, South Korea, and has been recently reported in various countries of Northern Europe, including Norway, Sweden, and Finland [16,17,18,19]. By far the country most affected by CWD is the USA. CWD prevalence is highly variable in North America, but can reach 30% for wild populations in endemic areas, and in exceptional cases can reach 80–90% in captive populations [7,10,20]. Over 200 captive deer facilities have detected CWD on their premises [20].

## 2. Mechanisms and Routes Responsible for the Efficient Natural Spreading of CWD

The mechanisms and factors explaining the highly efficient transmission of CWD in nature are not completely understood. It has been shown that CWD prion infection can be transmitted vertically or horizontally. Vertical transmission from infected mothers to newborns has been reported in deer as well as in other animal TSEs, including prion diseases in cattle and sheep [21,22]. In the case of cervids, it was shown that CWD-positive muntjac dams produced a progeny in which 80% of fetuses were infected with CWD, providing strong evidence for in utero prion infection [21]. Maternal CWD infection also appears to result in a lower percentage of live-birth offspring [21]. Natural CWD transmission from cows to calves has also been demonstrated to contribute to the CWD epidemic in free-ranging elk in Colorado [23]. Furthermore, several studies have reported the presence of PrP^Sc^ in fetal, gestational, and reproductive tissues of CWD-infected deer [24,25,26].

Horizontal transmission is possibly the most efficient route of natural spreading and as such the biggest contributor to the rapid spread of CWD [7,27,28]. Horizontal transmission can occur by direct animal-to-animal contact or indirectly through environmental fomites. Epidemiological studies of natural CWD spreading in captive mule deer found that in cohabiting animals, horizontal transmission is highly efficient, with an estimated incidence of 89% in herds in which vertical transmission was excluded [28]. Direct horizontal transmission of the disease from a CWD-infected animal to a healthy cervid likely occurs during mating or fighting through contact with saliva, blood, or mucosal tissues where CWD prions have been detected [29,30,31]. The efficiency of this type of transmission might be exacerbated by lesions on oral mucous membranes, which are frequent in cervids. Hoover et al. showed that minor lingual abrasions substantially facilitate CWD transmission in transgenic mouse models of CWD transmission [32]. Although in general prions are not considered airborne pathogens, some studies have shown that CWD can be transmitted by aerosol exposure [33,34], raising the possibility that simple cohabitation of uninfected and infected animals may result in prion transmission.

Indirect horizontal transmission occurs when uninfected animals become in contact with environmental materials that were previously contaminated by prions released by infected animals. CWD is a disease with a high peripheral distribution of infectious prions and it has been shown that infected animals can excrete a relatively large quantity of prions into the environment through urine, feces, or saliva [2,29,30,35]. Even though the prion-infectivity titer in CWD excreta is low, the repetitive excretion and accumulation of these materials lead to massive release of infectious prions into the environment. Estimations of the amount of prions excreted through feces and urine conducted by infectivity bioassays and high-sensitivity amplification techniques have shown that over the course of the incubation period and clinical disease, an infected deer sheds a higher amount of prions through excreta compared to the level of infectious agent present in the brain at terminal stage of the disease [29,36,37,38,39]. These calculations indicate that per year an infected animal may release >10 mg of infectious prions through feces and urine. Taking into account the population of CWD-infected cervids in the USA (likely to be in the order of hundreds of thousands of animals), the environment in the country possibly receive Kg quantities of CWD prions each year from urine and feces alone. The putative role of excreta in CWD transmission is supported by an experiment in which 11% of mule deer exposed to excreta-contaminated paddocks became infected [40]. CWD prions have been also found in deer saliva during the course of the disease, even in the asymptomatic incubation period [41,42,43]. Cervids produce a large amount of saliva, which can contaminate foliage and soil. Nevertheless, the largest amount of infectious material deposited at one point into the environment occurs when sick animals die and decomposing carcases remain in soil or are taken up by plants, insects, or scavenger animals. This source was proven in an experiment in which healthy mule deer were cohoused with CWD decomposed carcases. The result of this study showed a 25% infection rate in these animals [40].

Compelling evidence indicates that once prions are released into the environment, they can accumulate in association with various natural elements of the environment and remain infectious for several years. Soil is likely the environmental element most likely to interact with prions released via feces, urine, saliva, or animal carcasses. Prions can tightly bind to soil and remain infectious [44,45,46,47,48,49,50,51]. Interestingly, it has been reported that certain soil components (e.g., montmorillonite) may even increase the prion infectivity titer by over two orders of magnitude [52]. It is also possible that rainwater may elute and spread prions from contaminated soil, potentially carrying them to other areas, including groundwater [53]. Interestingly, CWD prions were detected in water in a CWD endemic area [54].

The documented presence of CWD prions in soil suggest that organisms living in contact with soil (e.g., plants, earthworms, and insects), may also be exposed and participate in prion spreading. In this sense, we showed that grass plants effectively bind prions from CWD-infected brain and excreta [55]. Prions can persist for prolonged periods of time bound to living plants [55], and importantly, prions bound to plants are infectious to experimental animals upon oral inoculation [55]. Surprisingly, we found that plants can uptake prions from the soil and transport them to the stem and leaves [55]. We also recently showed that earthworms living in prion-contaminated soil may contribute to spread prions across the environment. An experiment mimicking the interaction of earthworms with soil previously exposed to infectious prions found that worms can bind, uptake, accumulate, and scatter infectious prions in the soil [56]. Earthworms carrying prions maintain infectious properties [56]. Finally, we and others have shown that prions can bind to many natural and manmade environmental surfaces, including rocks, wood, metals, glass, plastic, concrete, etc. [57,58,59]. These surfaces commonly found in areas endemic for CWD can efficiently bind prions, which remain highly infectious. Strikingly, casual indirect contact with the prion contaminated surface (e.g., licking, sniffing, or brushing alongside the surface) is enough to transfer prions from the surface to the animal leading to disease [57]. Altogether, these findings suggest that prion contamination and progressive accumulation in the environment play a major role in CWD natural spreading. In addition, it is possible that prions attached to elements of the environment may also contribute to modify some prion properties, including infectivity titer, strain features, and possibly the potential to infect other animal species or even humans.

## 3. CWD Strain Diversity

Akin to conventional micro-organisms, prions can adopt multiple strains in a single species of animals [60,61]. However, in contrast from other microbes where strain diversity depends on changes on the nucleic acids, in the case of prions, strains are thought to arise from different conformations of PrP^Sc^ [60]. Prion strains can produce diseases with distinct clinical phenotypes, neurodegeneration profiles, incubation periods, areas of the brain affected by vacuolation and accumulation of PrP^Sc^, and biochemical/biophysical properties of PrP^Sc^. In humans affected by sCJD, five different prion strains have been recognized [62]. In animals, diverse prion strains have been reported in cattle and sheep, as well as in experimental rodents [63]. For CWD, we still do not know how many natural strains are present in nature [16]. Earlier studies from Telling et al. using transgenic mice models showed the existence of at least two different prion strains in CWD infected animals [64]. It was shown that host factors can modulate CWD prion diversity [65]. In addition, prion protein polymorphisms (Table 1), as well as small differences on the amino-acid sequence between different cervid species, may lead to the emergence of new prion strains when animals are infected with PrP^Sc^ carrying a mismatch sequence from host PrP^C^ [66,67,68,69].

It is well-established that the presence of certain polymorphic variants in cervids (Table 1) may confer resistance to CWD infection [18,70,71]. For example, infection of white-tailed deer harboring rarer polymorphisms at position 95 and 96 of PrP^C^ led to a prolonged incubation period compared to animals with more prevalent polymorphisms [72,73]. Supporting this conclusion, it was shown that wild white-tailed deer harboring the S96 polymorphism were less common in naturally CWD-infected animals [74]. Similarly, captive white-tailed deer containing uncommon polymorphisms (e.g., H95, S96, A116, or K226) have been shown to have a lower rate of CWD infection than those animals containing the more common polymorphisms [75]. Similar findings were observed in elk and mule deer, where the L132 and F225 polymorphisms, respectively, are underrepresented in CWD-infected animals [76,77]. Given the relatively large number of CWD-infected cervids in some areas of North America, it is possible that the differential susceptibility to CWD infection in certain polymorphic variants may alter the distribution of *Prnp* polymorphisms in cervids [78]. However, it is important to highlight that up to now, no polymorphisms confer absolute protection to CWD. Furthermore, it is likely that new emergent CWD strains may have a distinct preferred host selection than the commonly existing strains.

The recent emergence of CWD in Northern Europe has provided additional evidence for strain diversity in cervids. European CWD appears to have a different origin from North American CWD, since no cervids have been imported into Scandinavia from North America in the past several decades [18]. The first cases of CWD in Europe were reported in 2016–2019 by Benestad et al. in reindeer, moose, and red deer native to Norway [79,80,81]. In subsequent years, CWD was reported in moose living in Finland and Sweden [18]. Experiments by infectivity bioassay in bank voles provided strong evidence for European CWD corresponding to different strains from their North American counterparts [82]. This conclusion was further supported by infectivity studies in transgenic and gene-targeted mice expressing deer *Prnp*, showing convincing prion strain differences between European and North American CWD isolates [83]. Interestingly, repetitive passage in mice of some of the Norwegian CWD infectious material led to the adaptation and maturation of the prion strain to reach a similar stage as the current North America CWD strains [83]. This intriguing result suggests that European CWD is probably at a more primitive state of maturation than North American CWD. It has been shown that prions can evolve, mutate, adapt, and mature upon successive passages, leading to strain changes and resulting in the emergence of prions with different virulence and ability to cross species barriers [66]. Indeed, we previously showed that maturation of the CWD agent by many cycles of prion replication leads to increase ability to infect human models [84]. The strain diversity observed in CWD raises the possibility that distinct strains may have differential abilities to generate infection into other mammal species. Our studies using in vitro prion replication showed that Norwegian CWD has a higher potential to convert PrP^C^ from various animal species than North America CWD, but a lower potential to convert human PrP^C^ [85].

## 4. Species Barrier and CWD Zoonotic Potential

Another typical characteristic of prions in common with conventional micro-organisms is their capacity to infect only some animal species, a phenomenon usually referred as “species barrier” [86,87]. The ability of prions from one species to infect animals from a different species depends on the sequence homology between the prion protein from donor and acceptor as well as the specific prion strain properties of the infectious material [86,87]. Cervids cohabit in nature with several animal species susceptible to prion infection, such as livestock, rodents, scavengers, carnivores, pets, and humans (Figure 2) [7]. Particularly at risk are predators and scavenger animals, which may consume CWD-infected carcasses, but also herbivores through exposure to contaminated environment, including plants and soil. Several studies of experimental infection with CWD prions showed that CWD can transmit disease to various animal species, including cattle, sheep, goats, ferrets, and raccoons [88,89,90,91,92,93,94]. Interspecies transmission typically results in a lower attack rate (proportion of animals developing the disease out of the total number of animals exposed) and longer incubation periods compared to CWD transmission to cervids. Importantly, CWD transmission to other animal species may increase the virulence and zoonotic potential of newly generated prion strains in the host animals [2]. This appears to be the case of scrapie passaged into cattle. While traditional scrapie in sheep is considered not infectious to humans, cattle infected by scrapie can infect humans producing variant CJD (vCJD) [95].

In reference to biomedical relevance, the most important species barrier is the cervid-to-human barrier; in other words, the zoonotic potential of CWD prions. Many studies have been conducted using different technologies and model systems to assess the possibility that CWD may be able to infect humans under certain conditions. Currently, the evidence for transmission of CWD to humans is contentious. Studies using transgenic mice expressing human PrP^C^ injected with CWD prions have usually produced negative results, suggesting the species barriers between human and cervids is very large [96,97,98,99]. On the contrary, infectivity experiments in squirrel monkeys showed that CWD is highly infectious in this nonhuman primate model [100,101]. However, experiments in cynomolgus macaques, which are considered evolutionarily closer than squirrel monkeys to humans, have produced contradictory results. Two published studies from Chesebro’s group reported no clinical, pathological, or biochemical evidence of CWD transmission to macaques [101,102]. Studies were conducted in seven macaques inoculated with CWD-and sacrificed 11 to 13 years after CWD infection. Conversely, a yet-unpublished study by Czub, Schaetzl et al. found that upon infection of groups of macaques with the CWD agent by various routes, including oral inoculation of muscle tissue from infected cervids, many of the animals developed clinical, neuropathological, and biochemical alterations consistent with prion disease [103]. Analysis of the cervid-to-human species barrier by in vitro prion replication using the protein-misfolding cyclic-amplification assay (PMCA) or real-time quacking-induced conversion (RT-QuIC) indicated that under certain conditions, PrP^Sc^ from CWD-infected cervids is able to convert the human PrP^C^ into the pathogenic form [84,85,104,105,106]. Interestingly, the efficiency of crossing the cervid/human species barrier in vitro depended on the *Prnp* polymorphisms of the agent and the host [104] as well as on the degree of strain stabilization [84]. It has been shown that repetitive amplification of CWD prions as expenses of cervid PrP^C^ resulted in a PrP^Sc^ form more capable to convert human PrP^C^ [84]. These findings suggest that the longer animals are propagating CWD prions, the easier it might be to infect humans. Nevertheless, up to now there has not been any evidence that CWD has been naturally transmitted to humans. Epidemiological studies did not find a correlation between incidence of human TSE and CWD prevalence or consumption of deer meat [107,108].

Up to now, the only animal prion disease that has been transmitted to humans is BSE in cattle [95]. When compared to BSE, CWD appears to have a wider peripheral distribution of prions and much more efficient horizontal transmission. CWD prions have been readily detected in many peripheral tissues and biological fluids, including muscle, skin, lymph nodes, salivary glands, urinary bladder, pancreas, kidney, intestine, blood, urine, feces, and saliva [19]. This makes CWD prions more accessible to exposure to other animals and humans than BSE prions.

## 5. Strategies for Minimizing CWD Spreading: Is Eradication of CWD a Possibility?

Given the large number of CWD-infected animals in North America, the likely massive contamination of the environment, and the efficient horizontal spreading of the disease, it is unlikely that CWD can be eradicated from the United States. Nevertheless, it is possible to implement strategies to minimize and control CWD transmission. Following is an outline of six different strategies that may contribute to achieve this goal (Figure 3).

### 5.1. Surveillance

Routine active surveillance for CWD-infected cervids and for putative cases of CWD transmission to other animals or humans may contribute to limit the further spreading of the disease. CWD surveillance includes monitoring the appearance of new CWD cases both in wild and captive animals, the geographical location of these cases, possible origins, and animal species affected. This strategy might be particularly useful to prevent spreading of CWD to geographical areas not currently affected by the disease. An efficient surveillance program would require a tight interaction among state and federal authorities, owners of deer farms, hunters, wildlife agencies, and scientists experienced in CWD biology.

### 5.2. Selective Breeding

As described above, several reports have confirmed that some *Prnp* polymorphisms confer relative resistance to CWD infection [18,70,71,72,73,74,75,76,77]. Thus, one strategy to minimize CWD spreading might be to perform selective breeding to produce animals with several polymorphic variants known to decrease infection. A recent study has shown that selective breeding might contribute to reduce CWD spreading [109]. A similar strategy was previously employed to attempt reducing prevalence of scrapie in sheep [110]. However, the emergence of atypical strains of scrapie, which transmitted efficiently to animals with *Prnp* polymorphisms resistant to classical scrapie, complicated this approach [110]. Perhaps a safer and more effective strategy would be to produce knockout cervids for *Prnp*. It has been demonstrated that elimination of the *Prnp* gene in mice confers complete resistance to all forms of prions [111]. Interestingly, a *Prnp* null cow was previously generated, which showed no detectable abnormality and was completely resistant to BSE prion infection [112].

### 5.3. Implementation of High-Sensitivity Tests for CWD Prion Detection

Availability of a noninvasive test to detect live CWD-infected animals before the onset of the clinical disease would certainly help to minimize CWD spreading by identifying infected deer before they can excrete too many prions into the environment or transmit the disease horizontally or vertically. Currently, postmortem analysis of the brain or lymphoid tissues for vacuolation and PrP^Sc^ deposition by histology or by Western blot or ELISA techniques are the only ways to definitively diagnose CWD [18]. These assays have a rather low sensitivity and cannot be performed in live animals. In recent years, in vitro prion-replication assays, including PMCA and RT-QuIC, have been extensively used to detect prions in biological fluids of different animal species, including CWD [113,114,115,116,117,118]. These assays rely on mimicking in vitro the mechanism of prion replication to amplify minute amounts of PrP^Sc^ at expenses of large quantities of PrP^C^ using a cyclic-amplification procedure involving cycles of incubation and fragmentation. Both PMCA and RT-QuIC have shown high-sensitivity detection of CWD prions in a variety of biological samples, including blood, urine, saliva, and feces [31,37,39,41,43,119,120,121,122]. These assays have been also shown to be useful to detect prions in diverse environmental materials, including soil, plants, water, diverse natural and man-made surfaces, and in organisms living in the environment [47,54,55,56,57]. However, since they are not regularly performed, it is unclear how they will operate for routine prion detection.

### 5.4. Selective Animal Culling

Early removal of CWD-infected animals or cervids at a high risk for infection might be a possibility to decrease CWD prevalence [123,124]. Selective animal culling coupled with massive live animal testing with highly efficient assays for detection of CWD prions may represent a feasible strategy. For example, a study involving selective culling of CWD-infected mule deer (determined by immunohistochemistry of tonsil biopsies) in a limited area of Colorado showed a reduction of CWD prevalence in males [125]. However, other studies have found no evidence of CWD reduction after animal culling [126]. A massive nonselective culling of the entire population of deer in the Nordfjella part of Norway (>2000 animals) was carried out in 2017 after the emergence of CWD in this region [127]. This approach may well work to prevent new cases of the disease, especially at the beginning of the CWD epidemic, but it has serious ethical, economic, and political consequences.

### 5.5. Prion Decontamination

Although prions are resistant to many of the decontamination procedures utilized for conventional micro-organisms, there are very well-established chemical procedures for prion elimination [128]. The best-established procedures include treatment with solutions of concentrated sodium hypochlorite, sodium hydroxide, or guanidine. Unfortunately, some of these procedures are corrosive or not compatible with some surfaces. There are several reports of other prion-decontamination procedures (e.g., proteases, detergents, hydrogen peroxide) that can be less damaging to materials [129,130,131,132,133,134]. Coupling an efficient and noninvasive prion-detection methodology with an effective decontamination procedure could be very important to decrease further indirect horizontal spreading of the disease [2]. It is important to highlight that effort should be made to test the efficiency of decontamination in “real-life” treatments, and not assume the procedures work based on experimental studies. Fortunately, with the emergence of ultrasensitive prion-detection techniques (e.g., PMCA or RT-QuIC) that work in a variety of surfaces [57,135,136,137,138], it is now feasible to quality control the effectiveness of prion decontamination.

### 5.6. Treatment for CWD

Unfortunately, at this time there is no treatment available for CWD or any other prion disease [139,140]. Various different approaches have been proposed and tested in vitro on cells or animal models, including small-molecule drugs, passive and active immunization, aptamers, peptides, and RNA-interference techniques [139,140,141]. Although some of these strategies delay the onset of prion disease, none of them completely prevent or reverse the disease. In the case of CWD, a prophylactic treatment that can make animals immune to prion infection, such as a vaccine, could be ideal for massive use in preventing CWD spreading. Various studies investigating the use of a vaccine for CWD [141,142] have been reported, but more studies are needed to assess the efficacy of such approach in the field. Surprisingly, treatment of elk with a vaccine targeting a YYR disease-specific epitope resulted in an acceleration of disease onset compared to controls [143].

## 6. Conclusions and Perspectives

CWD is currently the most dangerous prion disease, because it affects wild animals, spreads efficiently, and has a high rate of peripheral excretion of infectious agent and a long incubation period. The mechanisms implicated in the facile transmission of CWD are not completely understood, but likely involve substantial contamination of the environment [2,144]. Also unknown are the number and properties of prion strains implicated in natural CWD and their contribution to disease spreading. Finally, the zoonotic potential of CWD remains a dangerous enigma.

Some of the most relevant pending questions in relation to CWD are: Is CWD infectious to humans under certain conditions? What is the exact contribution of the environment to CWD spreading? How many natural CWD strains exist and what are they properties? Has CWD spread to other species of animals? What is the atomic resolution structure of PrP^Sc^ responsible for CWD? How CWD prions produce brain damage and disease? What are the most efficient strategies to minimize and control CWD transmission? Is it possible to cure and eradicate CWD?

Much more research at all levels needs to be conducted to properly combat this insidious disease and to avoid the emergence of new diseases.

## Figures and Tables

**Figure 1 viruses-14-01390-f001:**
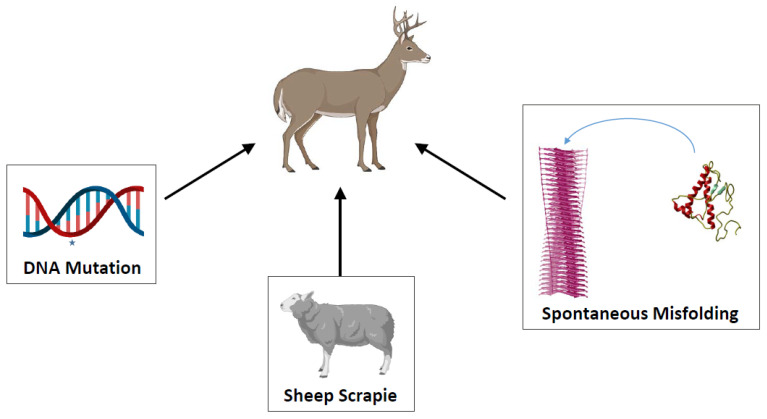
Schematic representation of the different hypotheses to explain the CWD origin. Although the origin of CWD is largely unknown, various hypotheses have been proposed, including transmission from a scrapie-infected sheep, a mutation in the *Prnp* gene (illustrated as a star in the left panel) and a spontaneous misfolding of PrP^C^ into PrP^Sc^.

**Figure 2 viruses-14-01390-f002:**
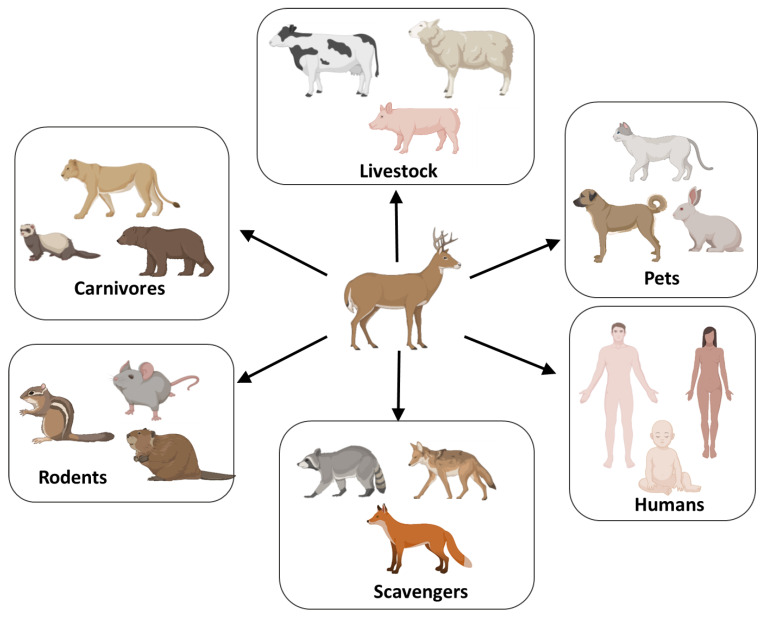
Animal species susceptible or at risk to be infected with CWD. Various animals cohabit with cervid, including livestock, rodents, scavengers, carnivores, domestic animals, and humans. Several studies of experimental infection of animals by CWD have shown that sheep, goats, cattle, ferrets, raccoons, and rodents can be infected with CWD [88,89,90,91,92,93,94].

**Figure 3 viruses-14-01390-f003:**
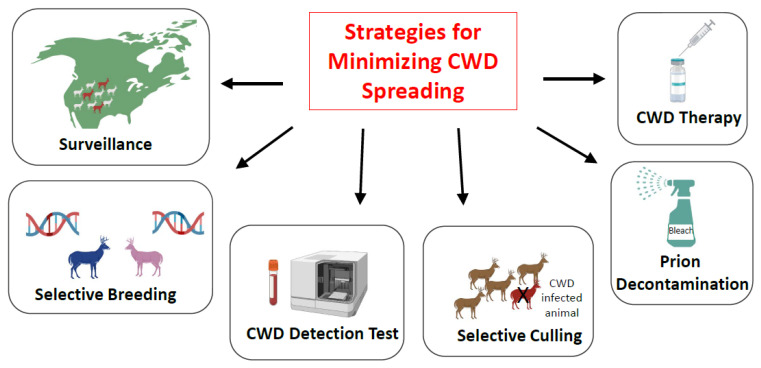
An overview of various strategies to minimize CWD spreading. Various alternative approaches have been proposed to decrease the risk of further CWD spreading, including detailed surveillance, selective breeding of animals harboring rare polymorphisms partially resistant to CWD, development of a highly efficient test for CWD detection, selective culling of infected animals, prion decontamination, and development of a treatment for CWD.

**Table 1 viruses-14-01390-t001:** Natural *Prnp* polymorphisms found in various species of cervids.

Cervid Species	Polymorphisms
White-tailed deer (*Odocoileus virginianus*)	G37V, G96S, G96R, A123T, Q230L
Mule deer (*Odocoileus hemionus*)	D20G, S225F, Q226K
Elk (*Cervus canadensis*)	M132L, E226
Reindeer (*Rangifer tarandus*)	V2M, Del 84-91, G129S, S138N, Y153F, V169M, N176D, S225Y, P242L
Red deer (*Cervus elaphus*)	G59S, T98A, P168S, M208I, Q226E, I247L
Moose (*Alces alces*)	T36N, S100R, K109Q, M209I

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
