# Peer review of "Transmission, Strain Diversity, and Zoonotic Potential of Chronic Wasting Disease"

_viruses, 2022, doi:10.3390/v14071390_

Round 1

Reviewer 1 Report

This review on CWD transmission, strain diversity and zoonotic potential is an excellent summary of the currently available knowledge on the subject.

I have one comment or question, which likely exceeds the scope of this specific review. Is it possible to compare BSE (which resulted in zoonosis) and CWD directly in terms of origin/strain diversity/infectivity of tissues and fluids/modes of transmission etc to explore the likely zoonotic potential of CWD?

Author Response

I would like to thank this reviewer for kind words about my article and useful suggestions. Below is a point-by-point answer to his/her comments

Comment: I have one comment or question, which likely exceeds the scope of this specific review. Is it possible to compare BSE (which resulted in zoonosis) and CWD directly in terms of origin/strain diversity/infectivity of tissues and fluids/modes of transmission etc to explore the likely zoonotic potential of CWD?

Response: This is a good suggestion. In the revised version, I included a paragraph about this. The origin of BSE and CWD are not entirely known, so we focused the discussion on the peripheral distribution of prions and modes of transmission.

Reviewer 2 Report

This review by Sandra Pritzkow describes Chronic wasting disease (CWD), a prion disease. The review describes the mechanism of spread, strain diversity, species barrier, and zoonotic potential. The strategies of minimizing CWD spreading is a new point added to the article compared to the previous review articles available. The review inspires the reader to think of developing new testing kits and treatment strategies for CWD.

Few minor points to be considered

1. The information on strain diversity due to polymorphism should be made more clear and informative by showing a table with all the polymorphisms reported to date. 

2. Citing this recent article will also be helpful. Hannaoui S, Triscott E, Duque Velásquez C, Chang SC, Arifin MI, Zemlyankina I, Tang X, Bollinger T, Wille H, McKenzie D, Gilch S. New and distinct chronic wasting disease strains associated with cervid polymorphism at codon 116 of the Prnp gene. PLoS Pathog. 2021 Jul 26;17(7):e1009795.

Author Response

I would like to thank this reviewer for kind words about my article and useful suggestions. Below is a point-by-point answer to his/her comments:

Comment: The information on strain diversity due to polymorphism should be made more clear and informative by showing a table with all the polymorphisms reported to date.

Response: Following the reviewer's advice we included a table with the known polymorphisms in cervids.

Comment: Citing this recent article will also be helpful. Hannaoui S, Triscott E, Duque Velásquez C, Chang SC, Arifin MI, Zemlyankina I, Tang X, Bollinger T, Wille H, McKenzie D, Gilch S. New and distinct chronic wasting disease strains associated with cervid polymorphism at codon 116 of the Prnp gene. PLoS Pathog. 2021 Jul 26;17(7):e1009795.

Response: I thanks the reviewer for this suggestion. The reference is now included.

Reviewer 3 Report

In this paper, Dr. Pritzkow provides an overview on the transmission, strain diversity and zoonotic potential of Chronic wasting disease.  The manuscript is well written, providing sufficient insight into the current knowledge and progress that has been made in understanding CWD.  Below are the minor points suggested for review.

Minor points:

1. Line 5 of the abstract, author states 'animals co-habiting with deer' - suggest changing deer to cervids

2. Background, paragraph 3, sentence 5 - remove 'to CWD' at the end of sentence (repetitive)

3. General comment: The term 'indeed' comes up repeatedly in close use throughout the paper.  Consider alternate terminology to make your point.

4. Conclusion: consider moving the bank of questions posed as concluding remarks to before the statement 'Much more research at all levels......'

Author Response

I would like to thank this reviewer for kind words about my article and useful suggestions. Below is a point-by-point answer to his/her comments

Comment: Line 5 of the abstract, author states 'animals co-habiting with deer' - suggest changing deer to cervids.

Response: Good point. Corrected

Comment: Background, paragraph 3, sentence 5 - remove 'to CWD' at the end of sentence (repetitive)

Response: Corrected

Comment: The term 'indeed' comes up repeatedly in close use throughout the paper.  Consider alternate terminology to make your point.

Response: I apologize for this repetition, which is now corrected.

Comment: Conclusion: consider moving the bank of questions posed as concluding remarks to before the statement 'Much more research at all levels

Response: Good point. I made the correction.

Reviewer 4 Report

Title: Transmission, strain diversity, and zoonotic potential of Chronic Wasting Disease

Authors: S. Pritzkow

Manuscript ID: viruses-1737936

Summary: In this manuscript, the author provides a background on prion disorders of both man and animals, focusing on chronic wasting disease (CWD) of cervid species.  The author reviews a bit of the biology and epidemiology before more in depth discussion on the mechanism and routes of CWD transmission, what is known about strain diversity for the prion agent, its species barrier and zoonotic potential, and strategies for slowing the spread of the disease and the potential for preventing infections in deer and elk species. 

Comments:    This manuscript is a good review of the above subject matter with regard to chronic wasting disease.  I have no suggestions on the material covered, though do have some minor comments that should be addressed prior to its final acceptance. 

Minor Comments:

1)      Throughout the manuscript, there are mostly minor grammar and usage mistakes that should be corrected, including overzealous use of commas, occasional incorrect terminology, and awkward English.  In the future, including line numbers in the submission may help point the author very quickly to these errors, but briefly:

a.     Third paragraph in the “Background” section: “…only prion disease in the wildlife…” should be “only prion disease in wildlife…”

b.      First paragraph of the “Mechanisms” topic: unnecessary comma in the sentence “…from infected mothers to newborns, has been reported…”

c.     First paragraph again of the same topic: elk females and young are referred to as cows and calves, not does and fawns as written.

d.    Third paragraph under the same topic: unnecessary comma in the sentence “…mule deer exposed to excreta-contaminated paddocks, became infected.”

e.   Third paragraph of the “Strain diversity” topic: “…moose and red deer inhabiting in Norway.” Should be “…moose and red deer native to Norway.” or similar.

f.     First paragraph under the “Species barrier” topic: livestock would fall under the category of domestic animals – perhaps the author means house pets?  It’s also not necessary to say “livestock animals,” when “livestock” would suffice.

g.  In the same paragraph, the author states that attack rate is the proportion of animals developing disease out of the total number of animals infected – it may be more appropriate to have the denominator be the total number of animals exposed or inoculated; infected implies they are diseased.

h.    At the end of this paragraph, the author states that cattle infected by scrapie can infect humans producing variant CJD without providing a reference. 

i.     In the second paragraph of the “Species barrier” topic: “…repetitive amplification of CWD prions as expenses of cervid PrP…” – I have no idea the message this sentence is meant to convey. 

                Other grammatical and usage mistakes are present throughout the manuscript, but it is beyond the scope of this review to correct an author’s grammar.  Having a colleague review the manuscript for additional errors is warranted. 

2)      In the fifth paragraph of the “Background” section, the second part of the paragraph seems overly repetitive and unnecessary – listing the upper limits of prevalence in captive populations twice, for example.

3)      In the second paragraph of the “Species barrier” section, the author may wish to reference a manuscript by K. Davenport et al. using RT-QuIC to probe the conversion efficiency of human PrP by CWD prions.  (J Virol 2015)

4)      Under section 5.4, Selective animal culling, the author may wish to reference a manuscript by N. Haley et al. using live animal testing and selective culling to manage CWD in farmed elk, which was unsuccessful.  (Prion 2020)

5)      Under section 5.6, CWD treatment, the author may wish to note that, to date, none of the various attempts to generate CWD vaccines have shown any evidence of success.  In fact, a publication by M. Wood et al. found that disease was accelerated in elk following vaccination. (Vaccine 2018)

In summary, this manuscript is a good review of the transmission, strain diversity, and zoonotic potential of CWD.  The author may wish to have a colleague review the manuscript for grammar and usage errors, including those listed.  The author should also address the minor comments provided above regarding select references and other various details.  I appreciate the opportunity to review this manuscript and look forward to reviewing any modifications the authors make based on these comments and those of other reviewers. 

Author Response

I would like to thank this reviewer for kind words about my article and useful suggestions. Below is a point-by-point answer to his/her comments

Comments: Throughout the manuscript, there are mostly minor grammar and usage mistakes that should be corrected, including overzealous use of commas, occasional incorrect terminology, and awkward English. 

Response: I apologize for these mistakes. All of them were corrected as recommended by the reviewer.

Comment: In the fifth paragraph of the “Background” section, the second part of the paragraph seems overly repetitive and unnecessary – listing the upper limits of prevalence in captive populations twice, for example.

Response: Following the reviewer's comment we re-worded the paragraph to remove redundancy.

Comment: In the second paragraph of the “Species barrier” section, the author may wish to reference a manuscript by K. Davenport et al. using RT-QuIC to probe the conversion efficiency of human PrP by CWD prions.  (J Virol 2015)

Response: The reference was included

Comment: Under section 5.4, Selective animal culling, the author may wish to reference a manuscript by N. Haley et al. using live animal testing and selective culling to manage CWD in farmed elk, which was unsuccessful.  (Prion 2020)

Response: Good point. I now discussed this study and included the reference.

Comment: Under section 5.6, CWD treatment, the author may wish to note that, to date, none of the various attempts to generate CWD vaccines have shown any evidence of success.  In fact, a publication by M. Wood et al. found that disease was accelerated in elk following vaccination. (Vaccine 2018).

Response: Many thanks to the reviewer for pointing out this interesting study. I now discuss briefly this study and included the reference.

Reviewer 5 Report

Review manuscript titled:  Transmission, strain diversity and zoonotic potential of Chronic Wasting Disease

Manuscript ID: viruses-1737936

Summary: This manuscript is a review that discusses about what is currently known and what is not about CWD transmission, strain diversity, and zoonotic potential. The review also has a section that shows some of the strategies that could be used to minimize CWD spreading.

Comments:

1.    Background, first paragraph: The term CJD must be introduced

2.    Background, third paragraph: I would recommend soften the language when stating that CWD origin, prevalence, and mechanisms of transmission remain incompletely understood.

3.    Background, fourth paragraph, line 3: “that” instead of “than”

4.    Figure 1, legend: “Prnp” instead of “Prnd”

5.    Background, fifth paragraph: some more recent references must be added when discussion about the places where CWD has been already detected. References 16 and 17 are from 2018 and 2017, respectively.

6.    Mechanisms and routes responsible for the efficient natural spreading of CWD, third paragraph: When stating that CWD infected animals can excrete a relatively large quantity of prions into the environment through urine, feces or saliva only one review reference was cited [2]. References from the original research showing those findings must be added.

7.    Mechanisms and routes responsible for the efficient natural spreading of CWD, third paragraph: References must be added on the sentences that discuss about the quantity of prions excreted by feces and urine

8.    Mechanisms and routes responsible for the efficient natural spreading of CWD, third paragraph: It is not clear why among excretory or secretory bodily fluids, saliva is possibly the one with the highest prion content. Is the amount of prions proteins detected in saliva higher than what can be detected in feces and urine? Any references that could support that sentence?

9.    Mechanisms and routes responsible for the efficient natural spreading of CWD, fourth paragraph: Could author be more specific when stating “a prolonged time”?

10. CWD strain diversity, first paragraph: References are missing on the first part of the paragraph (lines 1 to 6).

11. CWD strain diversity, first paragraph: sCJD was already introduced in the first paragraph of the background

12. CWD strain diversity, second paragraph: It would be helpful for readers if polymorphisms would be added instead of just saying “rarer”, “more prevalent”, and “more common” polymorphisms

13. Species barrier and CWD zoonotic potential, first paragraph: a reference is missing on the last two sentences.

14.  Figure 2, legend: Reference is missing

15. Strategies for minimizing CWD spreading: Is eradication of CWD a possibility?, Surveillance: In my opinion, this section is missing information. It would be interesting to provide more information about the current surveillance methods being used for CWD.

16.  Strategies for minimizing CWD spreading: Is eradication of CWD a possibility?, Is there any studies done for CWD specifically? It might be interesting to take a look on the following paper: https://www.mdpi.com/1265254

17. Strategies for minimizing CWD spreading: Is eradication of CWD a possibility?, Implementation of high sensitive tests for CWD prion detection: Based on the results shown on recent publications, I would suggest soften the language when discussing about the sensitivity of ELISA and immunohistochemistry for CWD detection. See: https://doi.org/10.3389/fvets.2021.824815 and https://doi.org/10.1128/JCM.01258-14

18. Strategies for minimizing CWD spreading: Is eradication of CWD a possibility?, Implementation of high sensitive tests for CWD prion detection: I would suggest softening the language in the sentence: “Both PMCA and RT-QuIC have shown high sensitive detection of CWD prions in a variety of biological samples, including blood, urine, saliva and feces”. While infectious CWD prions have been detected in saliva, blood, urine, and feces, is there really any evidence that support the use of “high sensitive detection”?

Author Response

I would like to thank this reviewer for kind words about my article and useful suggestions. Below is a point-by-point answer to his/her comments

Comment: Background, first paragraph: The term CJD must be introduced

Response: Fixed.

Comment: Background, third paragraph: I would recommend soften the language when stating that CWD origin, prevalence, and mechanisms of transmission remain incompletely understood.

Response: I replaced "remain incompletely understood" for "are not entirely understood"

Comment: Background, fourth paragraph, line 3: “that” instead of “than”

Response: Fixed

Comment: Figure 1, legend: “Prnp” instead of “Prnd”

Response: Fixed

Comment: Background, fifth paragraph: some more recent references must be added when discussion about the places where CWD has been already detected. References 16 and 17 are from 2018 and 2017, respectively.

Response: I added two new references (from 2021 and 2022).

Comment: Mechanisms and routes responsible for the efficient natural spreading of CWD, third paragraph: When stating that CWD infected animals can excrete a relatively large quantity of prions into the environment through urine, feces or saliva only one review reference was cited [2]. References from the original research showing those findings must be added.

Response: As suggested, I added the original references for these studies.

Comment: Mechanisms and routes responsible for the efficient natural spreading of CWD, third paragraph: References must be added on the sentences that discuss about the quantity of prions excreted by feces and urine.

Response: I am not sure what the reviewer's refers since references are already included to support the discussions in this paragraph.

Comment: Mechanisms and routes responsible for the efficient natural spreading of CWD, third paragraph: It is not clear why among excretory or secretory bodily fluids, saliva is possibly the one with the highest prion content. Is the amount of prions proteins detected in saliva higher than what can be detected in feces and urine? Any references that could support that sentence?

Response: There are some references to support this statement, but it is a bit preliminary and unconfirmed. Thus I decide to remove the statement that saliva is possibly the one with the highest prion content.

Comment: Mechanisms and routes responsible for the efficient natural spreading of CWD, fourth paragraph: Could author be more specific when stating “a prolonged time”?

Response: We changed "prolonged time" for "several years". References to support this statement are indicated in the sentences below this one.

Comment: CWD strain diversity, first paragraph: References are missing on the first part of the paragraph (lines 1 to 6).

Response: Following the reviewer's advice I included two references in this section.

Comment: CWD strain diversity, first paragraph: sCJD was already introduced in the first paragraph of the background

Response: Fixed.

Comment: CWD strain diversity, second paragraph: It would be helpful for readers if polymorphisms would be added instead of just saying “rarer”, “more prevalent”, and “more common” polymorphisms.

Response: Following the advice of second reviewer, I added a new table listing the polymorphisms in cervids.

Comment: Species barrier and CWD zoonotic potential, first paragraph: a reference is missing on the last two sentences.

Response: Two new references were added in this section.

Comment: Figure 2, legend: Reference is missing

Response: References were included in the text, but nevertheless, I added them to the figure legend.

Comment: Strategies for minimizing CWD spreading: Is eradication of CWD a possibility?, Surveillance: In my opinion, this section is missing information. It would be interesting to provide more information about the current surveillance methods being used for CWD.

Response: To expand this, I included a sentence stating what surveillance does.

Comment: Strategies for minimizing CWD spreading: Is eradication of CWD a possibility?, Is there any studies done for CWD specifically? It might be interesting to take a look on the following paper: https://www.mdpi.com/1265254

Response: Many thanks for pointing to this interesting study. I included a brief description of this paper and included the reference.

Comment: Strategies for minimizing CWD spreading: Is eradication of CWD a possibility?, Implementation of high sensitive tests for CWD prion detection: Based on the results shown on recent publications, I would suggest soften the language when discussing about the sensitivity of ELISA and immunohistochemistry for CWD detection. See: https://doi.org/10.3389/fvets.2021.824815 and https://doi.org/10.1128/JCM.01258-14.

Response: Following this reviewer advice, I soften the language discussing sensitivity of ELISA and IHC. 

Comment: Strategies for minimizing CWD spreading: Is eradication of CWD a possibility?, Implementation of high sensitive tests for CWD prion detection: I would suggest softening the language in the sentence: “Both PMCA and RT-QuIC have shown high sensitive detection of CWD prions in a variety of biological samples, including blood, urine, saliva and feces”. While infectious CWD prions have been detected in saliva, blood, urine, and feces, is there really any evidence that support the use of “high sensitive detection”?

Response: To soften the description of the potential use of PMCA and RT-QuIC, I added the following sentence "However, since they are not regularly performed, it is unclear how they will operate for routine prion detection"